# Molecular architecture and regulation of BCL10-MALT1 filaments

Florian Schlauderer[1], Thomas Seeholzer[2], Ambroise Desfosses[3], Torben Gehring[2], Mike Strauss [4], Karl-Peter Hopfner [1], Irina Gutsche[3], Daniel Krappmann[2] & Katja Lammens[1]

The CARD11-BCL10-MALT1 (CBM) complex triggers the adaptive immune response in lymphocytes and lymphoma cells. CARD11/CARMA1 acts as a molecular seed inducing BCL10 filaments, but the integration of MALT1 and the assembly of a functional CBM complex has remained elusive. Using cryo-EM we solved the helical structure of the BCL10-MALT1 filament. The structural model of the filament core solved at 4.9 Å resolution identified the interface between the N-terminal MALT1 DD and the BCL10 caspase recruitment domain. The C-terminal MALT1 Ig and paracaspase domains protrude from this core to orchestrate binding of mediators and substrates at the filament periphery. Mutagenesis studies support the importance of the identified BCL10-MALT1 interface for CBM complex assembly, MALT1 protease activation and NF-κB signaling in Jurkat and primary CD4 T-cells. Collectively, we present a model for the assembly and architecture of the CBM signaling complex and how it functions as a signaling hub in T-lymphocytes.

[1] Gene Center, Ludwig-Maximilians University, Feodor-Lynen-Str. 25, 81377 München, Germany. [2] Research Unit Cellular Signal Integration, Institute of Molecular Toxicology and Pharmacology, Helmholtz-Zentrum München - German Research Center for Environmental Health, Ingolstaedter Landstrasse 1, 85764 Neuherberg, Germany. [3] University Grenoble Alpes, CNRS, CEA, Institut de Biologie Structurale IBS, F-38044 Grenoble, France. [4] Department of Anatomy and Cell Biology, McGill University, Montreal, Canada H3A 0C7. These authors contributed equally: Florian Schlauderer, Thomas Seeholzer, Ambroise Desfosses. Correspondence and requests for materials should be addressed to I.G. (email: irina.gutsche@ibs.fr) or to D.K. (email: daniel.krappmann@helmholtz-muenchen.de) or to K.L. (email: klammens@genzentrum.lmu.de)

The immune stimulation of B-cell and T-cell receptors (TCR and BCR), as well as activating natural killer cell and fungal recognition receptors triggers activation of distinct Caspase recruitment domain (CARD)-containing scaffold proteins, including CARD9, CARD10 (also known as CARMA3), CARD11 (CARMA1), and CARD14 (CARMA2)[1,2]. CARD11, B cell lymphoma 10 (BCL10) and mucosa-associated lymphoid tissue lymphoma translocation protein 1 (MALT1) assemble the CARD11-BCL10-MALT1 (CBM) complex that bridges TCR/BCR proximal signaling to the canonical IκB kinase (IKK)/NF-κB and JNK pathway in lymphocytes[3]. Upon assembly, the CBM complex serves as a scaffolding platform that activates downstream signaling events via association of mediators such as ubiquitin ligases (e.g., TRAF6) and protein kinases (e.g., TAK1 and IKKβ)[4–6]. Upon activation, IKKβ phosphorylates IκBα leading to its proteasomal degradation and subsequent release, nuclear translocation and transcriptional activation of NF-κB[7]. Beyond the scaffolding function, MALT1 is a paracaspase and contributes proteolytic activity to the CBM complex, which is key for optimal lymphocyte activation and differentiation[8–10]. Altogether, CBM complex downstream pathways play important roles in regulating the activation, proliferation, and effector functions of lymphocytes in adaptive immunity[11].

The pathological relevance of CARD11, BCL10, and MALT1 is demonstrated by germline loss-of-function mutations associated with combined immunodeficiency (CID)[12–17]. In contrast, activating mutations in CARD11 promote B-cell proliferation and are frequently found in the malignant activated B-cell-subtype of diffuse large B-cell lymphoma (ABC DLBCL)[18–20]. Furthermore, chromosomal translocations leading to overexpression of BCL10 or MALT1 as well as generation of the API2-MALT1 fusion protein result in oncogenic activation associated with MALT lymphoma[21–23]. The clinical impact of the CBM signalosome has been emphasized by the discovery of MALT1 protease inhibitors that suppress antigen responses in T-cells and kill ABC DLBCL cells that arise from chronic BCR signaling[24–26].

In resting T-cells CARD11 is kept in an inactive conformation that is activated through phosphorylation by protein kinases including PKCθ and PKCβ[27,28]. In the hyper-phosphorylated conformation BCL10-MALT1 complexes are recruited via heterotypic interaction of the CARD11 and BCL10 CARD domains[29,30]. CARD11 acts as a molecular seed that upon binding induces the assembly of BCL10 filaments in vitro and in cells[31,32]. While the cryo-EM structure of the BCL10 filaments has been recently determined[31], no structural information is available for the integration of MALT1 into the BCL10 filaments. MALT1 constitutively associates with BCL10 in vitro and in cells. Mutational analyses suggested that regions in the C-terminal Ser/Thr-rich region of BCL10 interact with the N-terminal death domain (DD) and the two Ig (immunoglobulin)-like domains (Ig1/Ig2) of MALT1[33–36]. Nevertheless, the detailed nature of the BCL10-MALT1 interface remains unresolved.

To gain further inside into the CBM complex assembly we determined the cryo-EM structure of the BCL10-MALT1 complex. Our data define the exact interfaces for BCL10 oligomerization as well as the interaction of BCL10 CARD and MALT1 DD. Reconstitution assays of either KO Jurkat T-cells or murine CD4 T-cells from MALT1$^{-/-}$ mice highlight the significance of the interaction sites for CBM complex formation and activation of all CBM downstream signaling events.

## Results

### Cryo-EM structure of the BCL10-MALT1 filament.
To provide structural information about the interaction of BCL10 with MALT1 we performed cryo-electron microscopy (cryo-EM) of the human BCL10 (full length)-MALT1 (T29-G722) complex (Fig. 1a). BCL10 and MALT1 were co-expressed in bacteria and formed a stoichiometric complex that was purified to near homogeneity (Supplementary Fig. 1a). A cleavable GST-fusion at the N-terminus of BCL10 prevented filament assembly during purification and proteolytic removal of the GST-tag initiated oligomerization to a particle size of ~100 nm as determined by dynamic light scattering (DLS) (Supplementary Fig. 1b). An in vitro MALT1 cleavage assay revealed that the BCL10-MALT1 complex possesses protease activity (Supplementary Fig. 1c). Visual inspection of cryo-EM images of the purified complex, underpinned by 2D classification and examination of the power spectra, revealed that BCL10-MALT1 assembles into flexible helical filaments of ~29 nm in diameter, with an ordered inner core of ~14 nm in diameter and a less defined periphery (Fig. 1b–d). To calculate the high-resolution structure of the filament interior, the data were processed while limiting the diameter to 21 nm. This analysis was performed on the very first image frames 2–7 (total dose 14 electrons/Å$^2$) to avoid sample and image quality deterioration due to radiation damage. This resulted in a 4.9 Å resolution map of BCL10 CARDs tightly decorated by MALT1 DD (Fig. 1e–g).

Our analysis revealed that BCL10-MALT1 is a left-handed helix with a per subunit rotation of 100.8° and a rise of 5.083 Å, resulting in 3.571 subunits of BCL10-MALT1 per helical turn. This helical arrangement agrees with the one observed for the BCL10 (residue 1–115) filament alone, thus indicating that the overall arrangement of the BCL10 CARD core is largely unaltered within the BCL10-MALT1 complex structure (Fig. 2a–f)[31,32]. The 4.9 Å resolution map of the inner filament part enabled us to build a pseudo atomic model of BCL10 residues 10–115 and MALT1 residues (30–121) by flexible fitting of a BCL10 homology model and a related crystal structure of MALT1 DD into the EM density (Fig. 1h, i). Interestingly, the structural comparison with the MALT1 DD X-ray structure (pdb ID: 2G7R) reveals that in the presented EM density helix α6 is kinked and not forming an extended helix as previously proposed[37]. The observed fold of the kinked MALT1 helix is similar to other CARD family members (Fig. 1i).

### BCL10 filament assembly is critical for CARD11 recruitment.
The BCL10 CARD structure is considerably stabilized in comparison to the NMR structure due to the extensive network of interactions within the BCL10-MALT1 filament assembly (Fig. 2). To obtain a clearer perception of potential rearrangements of BCL10 upon interaction with MALT1, we compared the model of the BCL10 filaments alone, with our model of BCL10 in the BCL10-MALT1 complex[31]. Thereby, we found differences specifically in the amino acid registry that led to altered assignments of key residues involved in BCL10-MALT1 interaction (Supplementary Fig. 2a). Based on the refinement statistics and the FSC (Fourier Shell Correlation) curves assessing the fit between the maps and the models, our model is consistent with both our and the published EM densities[31] and provides reasonable stereochemistry and geometry values (Supplementary Fig. 2b).

The identified BCL10 CARD-CARD interfaces, type I, II, and III as diagrammed in Fig. 2a–f have been previously reported and are shown here to illustrate the composition of the BCL10 core filament[31]. The individual residues involved in the three types of interactions, in the presented structure are highlighted in Fig. 2d–f. Whereas type I and type II interactions are interstrand contacts between the helical turns (Fig. 2c–e), the type III interface exhibits interactions in the helical-strand direction (Fig. 2f). All interfaces identified by the BCL10 filament alone are

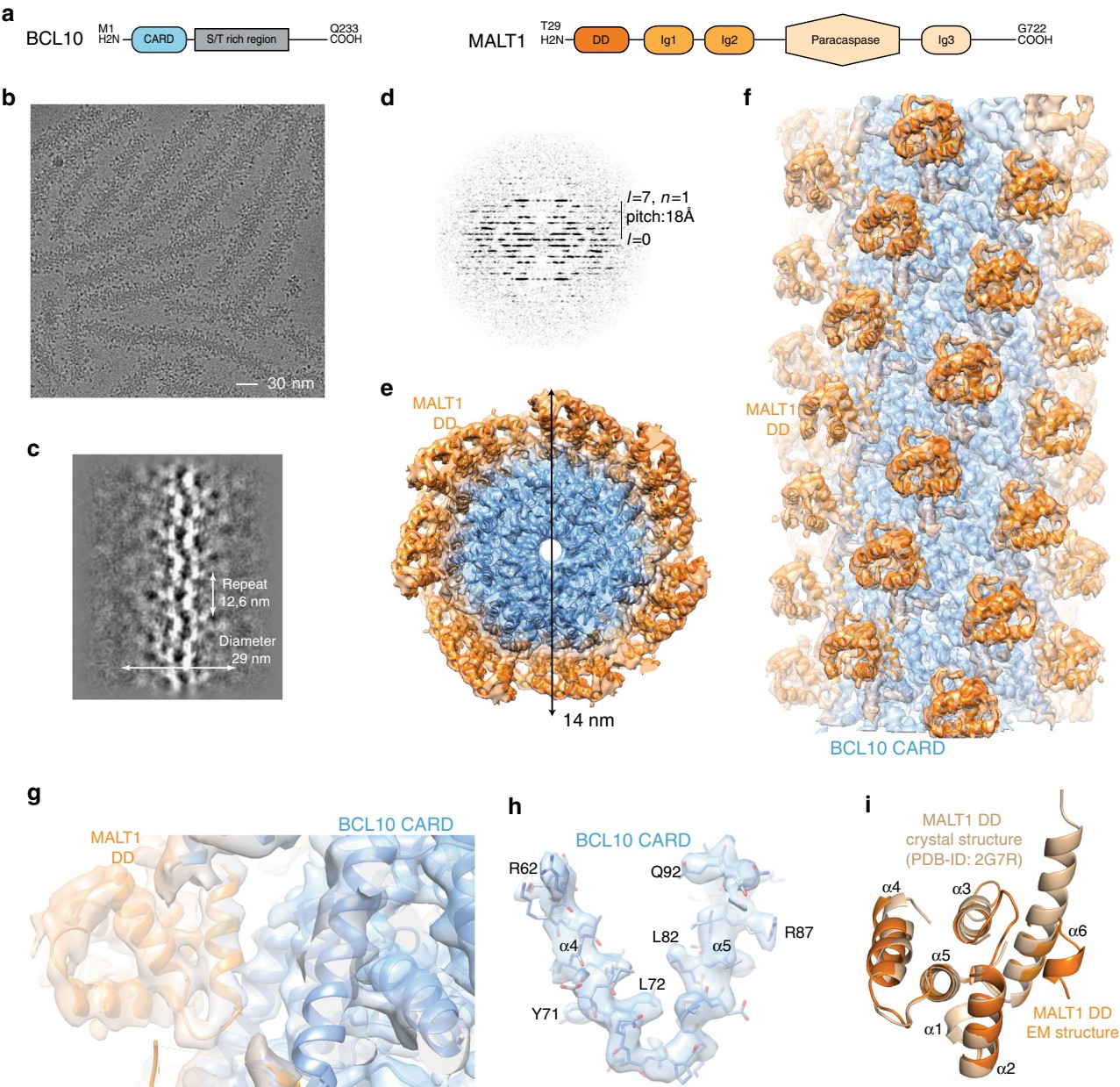

**Fig. 1** Cryo-EM reconstruction and atomic model of the BCL10-MALT1 filament. **a** Domain organization of the BCL10 and MALT1 protein constructs. **b** Example cryo-EM micrograph of the BCL10-MALT1 complex filaments. **c**, **d** 2D class averages and corresponding power spectra of BCL10-MALT1 filaments used for helical parameter determination. **e**, **f** Overall architecture of the BCL10-MALT1 filament assembly shown together with the cryo-EM density, clipped top and side view, respectively. The map shows the inner well-structured part of the filament at an overall resolution of 4.9 Å (FSC curve see Supplementary Fig. 8a). This part consists of the BCL10 CARD (residues 10–115) and the MALT1 DD (residues 30–121) colored blue and orange, respectively. **g**, **h** Example parts of the cryo EM density map shown together with the BCL10-MALT1 DD and the atomic model of BCL10 shown in **h** ribbon and stick representation, respectively. **i** Overlay of the MALT1 DD cryo-EM and crystal structure (pdb ID: 2g7r) colored orange and beige, respectively

largely conserved within the BCL10-MALT1 complex within the resolution limit of both structures[31,32] (Fig. 2d–f). Thereby, the type I interface comprises the most extensive interactions, with high electrostatic surface complementarity (Fig. 2d). Additionally, we identified residue R42 in interface I and R36 in interface II to be important for BCL10 filament assembly (Fig. 2d, e). To validate the importance of the interfaces we generated site-directed mutants and compared their capability to oligomerize in vitro with those of the wild-type complex. The BCL10 mutations R42E, R49E, and R36E abrogated the ability of the BCL10-MALT1 complex to oligomerize, while the interaction with MALT1 was retained as shown by DLS and co-purification,

respectively (Supplementary Fig. 2c, d). This emphasizes the relevance of the identified interaction regions for filament assembly but not MALT1 interaction.

Previously, the effect of BCL10 interface I-III mutants on NF-κB and MALT1 activation was analyzed upon overexpression[32]. Since filament assembly with recombinant purified BCL10-MALT1 is strongly influenced by local protein concentration, we aimed to investigate the biological effects of BCL10 oligomerization mutants in a clean genetic setting without overexpression or perturbation by endogenous BCL10. To this end we generated BCL10 KO Jurkat T-cells by CRISPR/Cas9 technology using sgRNA targeting Exon1 of human *BCL10*

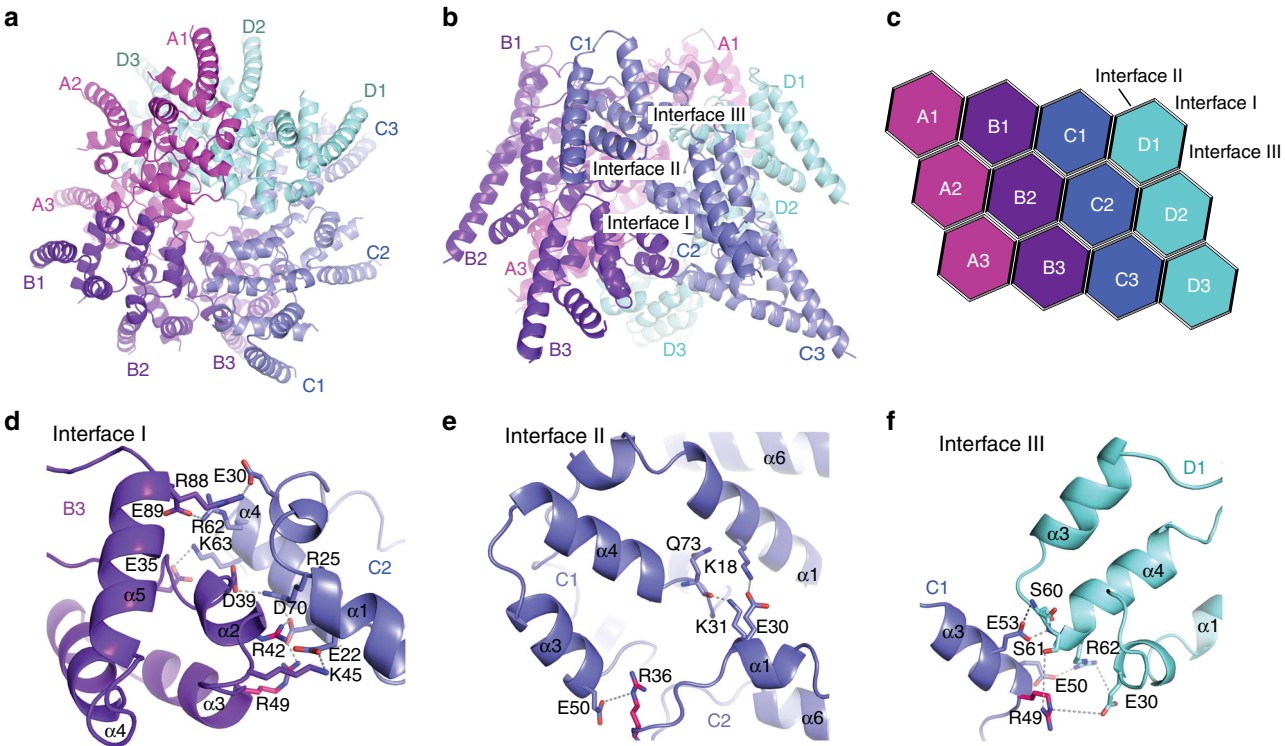

**Fig. 2** Core structure of the BCL10 CARD filament. **a**, **b** Top and side view of the BCL10 protein model obtained by interpretation of the BCL10-MALT1 complex cryo-EM density. **c** Schematic diagram of the BCL10 helical assembly. Each CARD is represented as a hexagon and the three interfaces are indicated accordingly. **d**, **f** Detailed view of the three BCL10 interaction surfaces I (**d**), II (**e**), and III (**f**). Interacting residues are shown in stick representation. Sites mutated R42, R36 are highlighted in magenta

(Supplementary Fig. 3a). After limiting dilution, we obtained a BCL10 KO Jurkat T-cell clone, which was reconstituted with BCL10 wt or the BCL10 interface I (R42E) and II (R36E) mutants, which lost the ability to assemble filaments in vitro. To exclude that the cellular effects are due to overall conformational alterations in the BCL10 CARD, by changing positively charged arginines to negatively charged glutamic acids, we included the less severe BCL10 R42A substitution (Fig. 3b). Lentiviral infection led to homogenous transduction of Jurkat T-cells as judged by co-expressed surface marker ΔCD2 and equivalent expression of all constructs close to endogenous levels in the BCL10 KO cells (Supplementary Fig. 3b–e). Precipitation of BCL10 either by co-IP or StrepTactin pull-down (ST-PD) demonstrated that the mutation in the BCL10 filament interfaces I and II prevented the PMA/Ionomycin (P/I) inducible recruitment of BCL10 to CARD11 and thus stimulus-dependent CBM complex formation (Fig. 3a, b). However, none of the mutants impaired constitutive binding of BCL10 to MALT1, suggesting that the overall protein structure is still intact.

To determine the functional relevance of the BCL10–BCL10 interfaces for T-cell signaling, we stimulated the Jurkat T-cells expressing the distinct BCL10 mutants and analyzed NF-κB signaling and MALT1 protease activation (Fig. 3c, d). Indeed, interface I mutations BCL10 R42E or R42A and interface II mutation R36E completely abolished IκBα phosphorylation and degradation as well as NF-κB DNA binding after P/I stimulation. Likewise, activation of the MALT1 protease was absent in all BCL10 filament mutants as determined by the ability to cleave the substrates BCL10, CYLD, and HOIL1 (Fig. 3c, d). Hence, under physiological conditions the correct assembly of the BCL10 core filament via its CARDs is crucial for stimulus-dependent CARD11 recruitment and thus CBM complex assembly and downstream functions.

**MALT1 binds to BCL10 via DD and CARD interaction**. Besides unravelling the architecture of the BCL10 CARD filaments, the 4.9 Å resolution cryo-EM map provides the detailed molecular structure of the BCL10-MALT1 interaction surface (Fig. 4a–c). The MALT1 DD interacts with BCL10 at the rim of the CARD core filament in a 1:1 stoichiometry. Thereby, the C-termini of MALT1 DD domains are pointing away from the core filament. A close-up view of the BCL10-MALT1 interaction site I (BM-I) illustrates that MALT1 binds to the C-terminal part (BCL10 helix α6) of the BCL10 CARD forming a new interface that is distinct from the BCL10 filament interfaces I–III (Fig. 4a–c). Thus, the structure underscores that BCL10 filament assembly is not required for MALT1 association, which agrees with the cellular pre-assembly of the BCL10-MALT1 complex without stimulation (see Fig. 3a, b). In BM-I, the two most central interacting residues of MALT1 are V81 and L82 situated in helix α4 (Fig. 4c). The opposing hydrophobic surface in BCL10 is formed by L104, V103 in helix α6 and V83 and I96 in helices α5 and α6, respectively. Several salt bridges stabilize the hydrophobic core contacts (Fig. 4c).

To investigate the significance of the identified BM-I, we mutated residues on both sides of the interface. Co-purification of GST-BCL10 demonstrates a severely reduced binding of MALT1 V81R mutant in vitro (Supplementary Fig. 4a). In line, the single mutation V81R completely disrupted the interaction of BCL10 after co-expression in HEK293 (Fig. 4d). Vice versa, BCL10 L104R abolished association of MALT1 to BCL10 (Fig. 4d). Further, the cryo-EM structure indicated a potential second interface between MALT1 DD α4 and α5 segment and the adjacent BCL10 helix α6 at the C-terminus of the core CARD filament (BM-II; Supplementary Fig. 4b, c). However, mutation of potential contact points Q76A/E98R as well as the putative disulfide bond C77-C91 (C77A) in MALT1 did not affect BCL10-

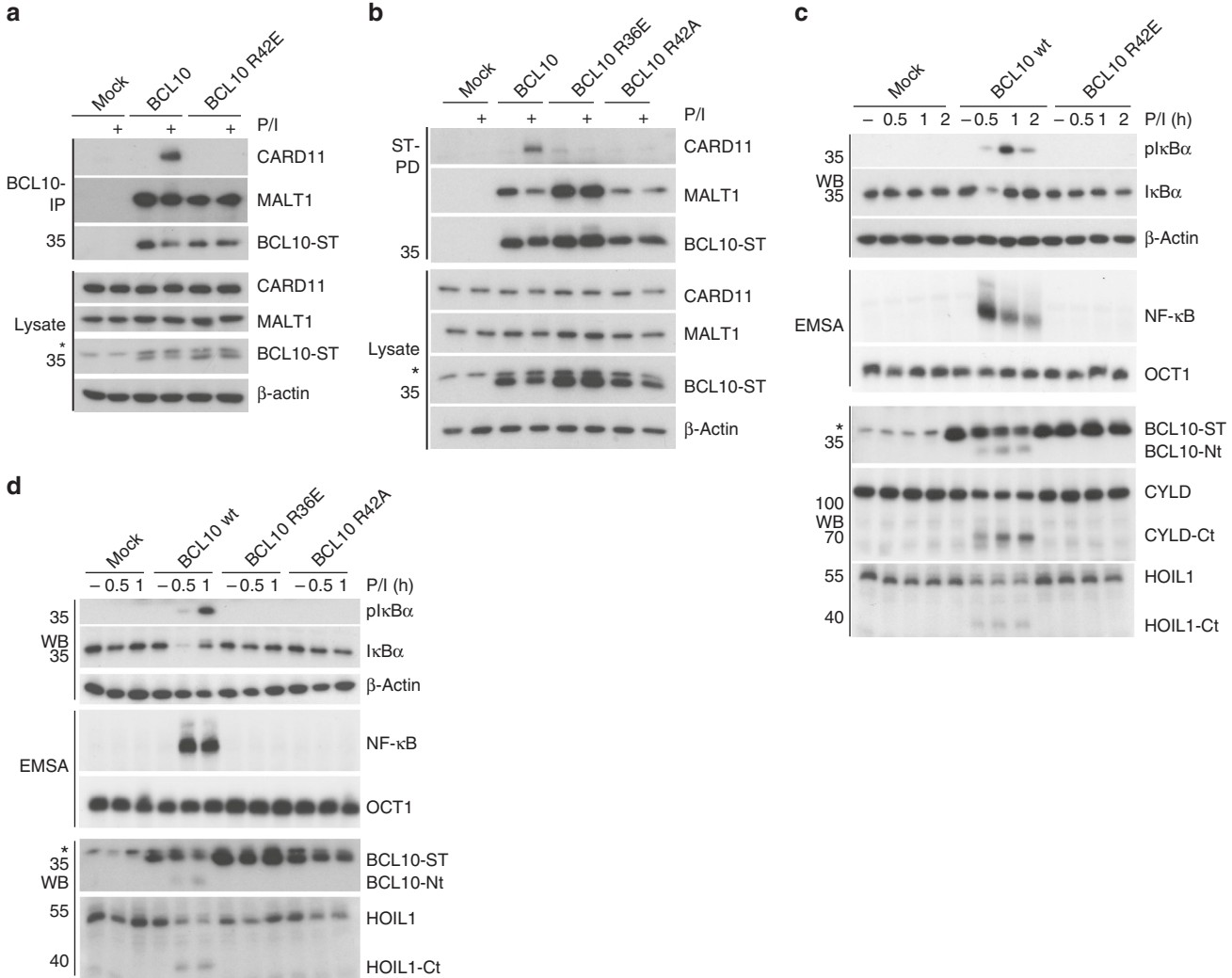

**Fig. 3** Functional analyses of BCL10-BCL10 interfaces in Jurkat T-cells. **a**, **b** BCL10 KO Jurkat T-cells lentivirally reconstituted with BCL10 wt, R42E, R36E or R42A constructs were stimulated P/I for 20 min and CBM complex formation was monitored by BCL10-IP (**a**) and Strep-tag II (ST) pull down (PD) (**b**). **c**, **d** BCL10 KO Jurkat T-cells reconstituted with BCL10 wt or BCL10-BCL10 interface mutants were stimulated with P/I for the times indicated. NF-κB activation was determined by IκBα WB and EMSA. Activation of the MALT1 protease was monitored by WB analyses of the substrate cleavage. Asterisks indicate non-specific cross-reactivity of the BCL10 antibody

MALT1 association in co-IPs (Supplementary Fig. 4d), suggesting that BM-II is not essential for BCL10-MALT1 complex formation.

**CBM assembly and T-cell activation relies on MALT1 binding.** To assess the biological impact of the newly identified BCL10-MALT1 interaction surfaces, we expressed BM-I mutants located on both sides of the interface in BCL10 or MALT1 KO T-cells, respectively (Fig. 5). MALT1 KO Jurkat T-cells were generated by CRISPR/Cas9 technology[38]. In BCL10 we used the single mutation L104R in helix α6 of the CARD. In MALT1 we analyzed mutation V81R or L82D in α4 of the DD opposite to the BCL10 α6 helix. Using lentiviral transduction, we obtained Jurkat T-cells that expressed all BCL10 or MALT1 constructs homogenously at endogenous levels (Supplementary Fig. 5a–f). Indeed, BCL10 co-IP experiments revealed that missense mutation either in BCL10 (L104R) or MALT1 (V81R or L82D) completely abrogate the constitutive binding of BCL10 to MALT1 in resting Jurkat T-cells (Fig. 5a–c). For BCL10 L104R and MALT1 V81R, loss of interaction was also confirmed by ST-PD of BCL10 or MALT1, respectively (Supplementary Fig. 5g-h). Moreover, loss of MALT1

led to a severely reduced BCL10 recruitment to CARD11 after P/I stimulation (Fig. 5b, c). In all BCL10 and MALT1 interface mutants the binding of BCL10 to CARD11 was impaired, providing evidence that the constitutive association of BCL10 and MALT1 is a prerequisite for CBM complex assembly (Fig. 5a–c). Thus, in T-cells the BCL10-MALT1 interaction significantly contributes to the dynamics of CBM complex formation strengthening the importance of interface BM-I.

To determine the functional relevance of the BCL10-MALT1 interface for T-cell signaling and MALT1 activation, we stimulated the Jurkat T-cells expressing the MALT1 and BCL10 BM-I mutants (Fig. 5d–h). In line with its critical role for assembly of the entire CBM complex, IκBα phosphorylation/ degradation and NF-κB DNA binding after T-cell stimulation were abolished. Also, activation of the MALT1 protease was absent in the BCL10 L104R (Fig. 5d, e) as well as MALT1 V81R (Fig. 5f, g) or MALT1 L82D (Fig. 5h) mutants, as evident from the lack of substrate cleavage. In contrast, BM-II mutations Q76A/E98R and C77A, which did not significantly impair BCL10-MALT1 association, did not alter NF-κB responses (Supplementary Fig. 4d and 5i). To address the importance of

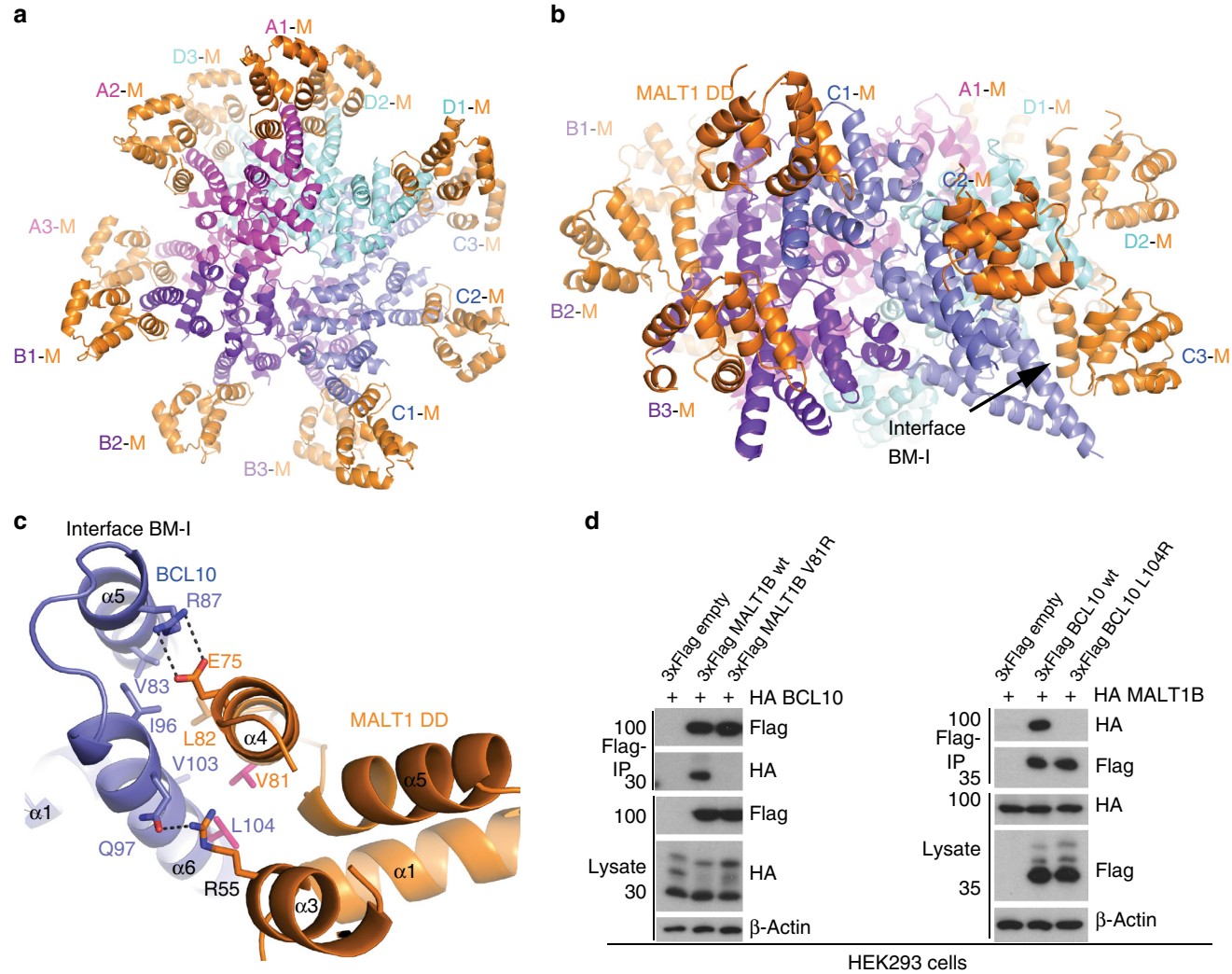

**Fig. 4** Architecture of the BCL10-MALT1 interface. **a**, **b** Top and side view of one repeat of the BCL10-MALT1 holo-complex as visible in the cryo-EM density. Position of BCL10-MALT1 interface BM-I is indicated in (**b**). **c** Close up view of the BCL10 and MALT1 interaction site I (BM-I) shown as ribbon model in blue and orange, respectively. The interacting residues are shown in stick representation. Mutations introduced are colored magenta. **d** HEK293 cells were co-transfected with tagged MALT1B and BCL10 wt and mutant constructs as indicated. Co-IP was carried out using anti-Flag antibodies and analyzed by WB for co-precipitation

the BCL10-MALT1 interface BM-I for T-cell activation in a physiological setting, we purified CD4 T-cells from MALT1$^{-/-}$ mice and reconstituted the cells with either MALT1 wt or the MALT1 BM-I mutant V81R. Effects on NF-κB signaling and IL-2 production after MALT1 reconstitution were determined by single cell FACS analyses. Retroviral transduction led to equal expression of MALT1 wt and the mutant construct V81R in primary T-cells (Fig. 5i). Infected cells were identified by the surface marker Thy1.1 (Supplementary Fig. 6a-b). As expected, IκBα was not degraded and IL-2 was not significantly induced in Thy1.1-negative MALT1$^{-/-}$ T-cells (Supplementary Fig. 6a-b). When gated on the transduced Thy1.1-positive CD4 T-cells population, MALT1 wt but not BM-I interface mutant V81R was able to rescue NF-κB signaling after P/I stimulation as evident from IκBα degradation (Fig. 5j). Further, strong upregulation of IL-2 in response to P/I stimulation or anti-CD3/CD28 co-ligation in CD4 T-cells was impaired in the MALT1 V81R mutant expressing cells as determined by intracellular FACS staining (Fig. 5k, l). Thus, the identified BCL10-MALT1 interface I is essential for bridging TCR stimulation to downstream signaling and T-cell activation.

**Architecture of the BCL10-MALT1 holo-complex.** To gain more information about the structural organisation of the additional MALT1 domains (Ig1-Ig2-paracaspase-Ig3) around the BCL10-MALT1 DD skeleton, the cryo EM data have been analysed by a second approach. The diameter constraints have been relieved to 29 nm and the dose increased to 32 electrons/Å$^2$ (frames 2–16) which yielded an overall 5.9 Å, but highly anisotropic resolution reconstruction of the entire BCL10-MALT1 complex domains (Fig. 6 and Supplementary Fig. 8). In the density filtered to 6 Å the MALT1 DDs are rigidly attached to the BCL10 core (Fig. 6a, b). The EM density filtered to 8 Å depicts that the subsequent MALT1 domain Ig1 is clearly visible and we were able to rigid body dock this domain. The Ig1 domain is pointing away from the BCL10-MALT1 DD core filament separating the C-terminal domains from the inner core of the filament (Fig. 6c, d). Together with the flexible linked Ig2 domain this arrangement supposedly allows for dimerization of the MALT1 paracaspase domains, which is key for MALT1 protease activation[39]. Further, the map with a resolution cut-off at 25 Å indicates how a stable inner core formed by BCL10 CARD and the MALT1 DD-Ig1 fragments orchestrates a flexible outer

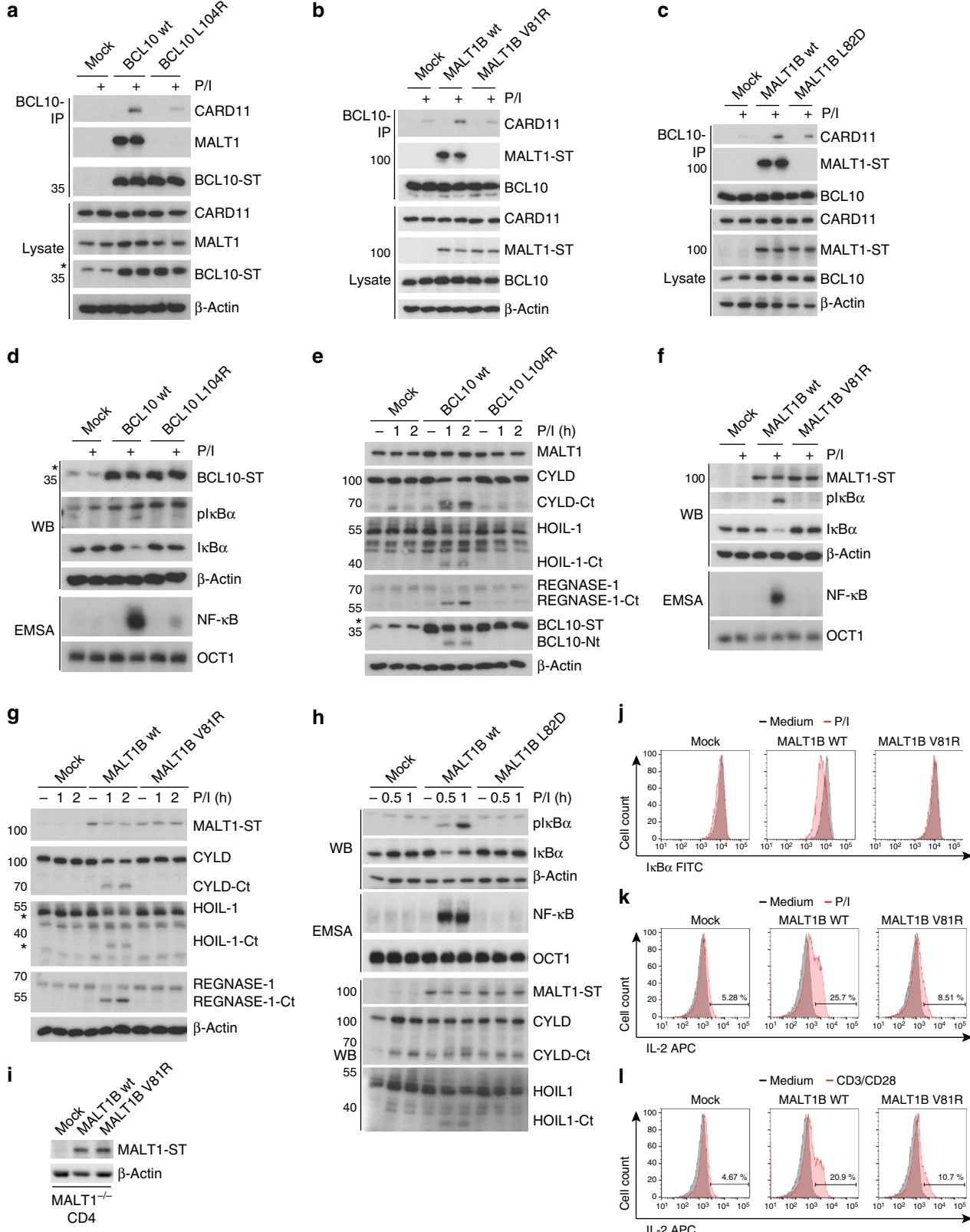

MALT1 platform (Fig. 6e, f). The top view illustrates a 'paddle wheel-like' architecture of the BCL10-MALT1 complex with the MALT1 Ig1-Ig2-paracaspase domains forming the more adaptable periphery of the filament (Fig. 6f). Due to the flexibility of the C-terminal MALT1 domains, we could not provide an unambiguous fit of the MALT1 Ig2-paracaspase-Ig3 domains.

## Discussion

Our cryo-EM structure resolved the inner core of the BCL10-MALT1 assembly at 4.9 Å resolution. We identified the BCL10 CARD and MALT1 DD interface BM-I and cell-based assays provide compelling proof that the filaments channel all CBM downstream signaling events. Since the peripheral C-terminal

**Fig. 5** Functional analyses of the MALT1-BCL10 interface in Jurkat and murine CD4 T-Cells. **a** BCL10 KO Jurkat T-cells were lentivirally reconstituted with BCL10 wt or BM-I mutant L104R constructs. After 20 min P/I stimulation CBM complex formation was investigated by BCL10-IP. Asterisk indicates non-specific cross-reactivity of the BCL10 antibody. **b**, **c** MALT1 KO Jurkat T-cells were lentivirally reconstituted with MALT1 wt or BM-I mutant constructs V81R (**b**) or L82D (**c**). P/I stimulation and CBM complex formation was investigated as in (a). **d–h** NF-κB signaling (**d**, **f**, **h**) and MALT1 protease activation (**e**, **g**, **h**) in BCL10 (**d**, **e**) or MALT1 (**f–h**) BM-I mutant reconstituted Jurkat T-cells after P/I treatment was analyzed by Western Blot (IκBα phosphorylation and degradation) and NF-κB-DNA binding studies (EMSA). Induction of MALT1 protease activity was monitored by cleavage of MALT1 substrates in WB as indicated. **i** Expression of MALT1 wt and BM-I mutant (V81R) was determined by WB after enrichment of infected murine MALT1$^{-/-}$ CD4 T-cells. **j** MALT1$^{-/-}$ CD4 T-cells transduced with MALT1 wt or BM-I mutant were stimulated for 30 min with P/I. IκBα expression and degradation were measured by FACS and transduced cells were gated by co-staining of Thy1.1 (Supplementary Fig. 6a). **k**, **l** MALT1$^{-/-}$ CD4 T-cells transduced as in (**j**) and stimulated for 5 h with P/I (**k**) or anti-CD3/CD28 (**l**). Intracellular IL-2 production was determined by FACS and transduced cells were gated by co-staining of Thy1.1 (Supplementary Fig. 6b)

regions of MALT1 and BCL10 are flexible, it is unlikely that other surfaces are directly contributing to BCL10 binding. This is remarkable, because previous co-IPs using overexpression suggested that BCL10-MALT1 may associate through a more extended binding surface. BCL10-MALT1 interaction was lost by a small deletion at the C-terminus of the CARD (aa 106–120), which overlaps the MALT1 binding surface that has been identified here by cryo-EM[34]. However, other deletions in the BCL10 C-terminus also impaired the tight association of BCL10 to MALT1[36]. Moreover, on the side of MALT1 it was shown that the DD as well as the Ig1/Ig2 domains are contributing to the BCL10 binding in cells[33]. Thus, we speculate that regions outside the direct BCL10-MALT1 interface are important for the conformation of the CARD and the DD and thus indirectly affect the interaction. In fact, the crystal structure of the MALT1 DD alone indicates that in the absence of the Ig1/Ig2 domains the α6 helix is not kinked, which could certainly prevent BCL10 from binding[37].

Further, we found that MALT1 association to BCL10 is required for the stimulus-dependent recruitment of BCL10 to CARD11. In vitro CARD11 and BCL10 form complexes at high concentrations[31,32]. Under physiological conditions, however, only the MALT1-BCL10 complex adopts a conformation that allows efficient CARD11 association. Mechanistically, MALT1 may either facilitate a BCL10 conformation that promotes CARD11 binding or wrapping of BCL10 by MALT1 may stabilize otherwise labile BCL10 core filaments. Clearly, the cryo-EM structure reveals stoichiometric binding of BCL10 to MALT1 and that BCL10 oligomerization is not required for MALT1 association, which is in line with the constitutive pre-assembly of BCL10-MALT1 complexes in cells. Furthermore, BCL10 is decorated with ubiquitin chains and these BCL10 modifications have been suggested to activate downstream signaling partially bypassing the necessity for MALT1[6,40]. However, the absence of a CARD11-BCL10 binding in MALT1 deficient or BM-I mutant cells indicates that signaling competent CARD11-BCL10 sub-complexes do not exist and downstream effects are relying on MALT1 association.

The cellular analyses of structure-guided mutations in the BCL10 filament interfaces I and II highlight the necessity of filament assembly for CARD11 recruitment. Since none of these mutations affect MALT1 association, it is quite unlikely that they destroy the overall conformation of the BCL10 CARD or the interface to CARD11. In line with the modeled CARD11-BCL10 interactions[32], we propose that the weak affinity of BCL10 monomers for CARD11 needs to be stabilized by BCL10 oligomerization, which augments the affinity by presenting multiple interfaces. In fact, the data implicate that recruitment of BCL10 to CARD11 and BCL10 oligomerization are highly interconnected processes that cannot be uncoupled. Thus, BCL10 oligomerization may boost an initial low affinity interaction leading to rapid CBM complex assembly and threshold responses after stimulation.

Furthermore, an interesting aspect regarding the BCL10-MALT1 core structure is, that BCL10 is degraded upon prolonged T-cell stimulation, leading to CBM complex disassembly and termination of TCR-signaling[41,42]. Proteasomal degradation requires unfolding of the proteins and thus the rigidity of BCL10 core filaments can explain why BCL10 is primarily removed through selective autophagy and lysosomal degradation[41,43]. However, despite the 1:1 stoichiometry of BCL10-MALT1 in the complex, MALT1 is stable and not degraded after stimulation[43]. At present it is unclear how MALT1 is disconnected from the BCL10 filaments. MALT1 auto-cleavage at R149 could certainly release the C-terminal fragment, but there is no evidence for the appearance of a stabilized DD-truncated form of MALT1 after stimulation[44]. The BCL10 C-terminus is most likely flexible and was not visible in the cryo-EM map, but interestingly hyperphosphorylation of BCL10 in this Ser/Thr-rich region impairs binding to MALT1[36]. Thus, post-translational modifications of BCL10 or MALT1 may be involved in the release of MALT1 to separate it from removal by autophagy.

Despite the lower resolution in the outer region of the 29 nm wide BCL10-MALT1 filaments, the cryo-EM map depicts how the C-terminal MALT1 domains are emanating from the BCL10-MALT1 DD core filament forming a 'paddle wheel-like' shape. In spite of the moderate resolution of the map of the outer regions in the BCL10-MALT1 complex, the current data allows us to provide a model in which MALT1 C-terminal region covering the Ig2-paracaspase-Ig3 domain is flexibly attached to the BCL10-MALT DD core filaments. The flexibility and positioning of the individual MALT1 molecules in the filament periphery provides a platform for the recruitment of mediators like TRAF6 to foster NF-κB signaling[38]. Further, mono-ubiquitination, substrate binding and paracaspase domain dimerization will promote protease activation[39,45]. Future studies must elucidate the mechanism of MALT1 protease activation in the complex and how the BCL10-MALT1 platform integrates factors like TRAF6, TAK1 and NEMO/IKKβ to initiate downstream processes.

## Methods

**Expression and purification**. Human MALT1 isoform B amino acids T29 to G722 and the V81R mutant of this construct were cloned by Nde1 and Not1 restriction sites into a modified pET28a vector (Novagen), containing a N-terminal PreScission Protease (GELifeSciences) cleavable 8 × -His-tag sequence (Supplementary Table 2). Human BCL10 wt and the respective R36E, R42E, R49E mutants were cloned into pGEX-6P-2 vector (GELifeSciences) using BamH1 and Xho1 restriction sites (Supplementary Table 3)[36]. The corresponding plasmids were co-transformed in *Escherichia coli* Rosetta™ (DE3) strain (Novagen), protein expression was induced by addition of 0.2 mM IPTG and performed overnight at 18 °C. Cells were resuspended in lysis buffer containing 50 mM Hepes pH = 7.5, 200 mM NaCl, 7 mM Imidazol and 4 mM β-mercaptoethanol. Cells were lysed by sonication and clarified by centrifugation. The BCL10-MALT1 complex was further purified by Ni-NTA affinity chromatography (Qiagen) using the lysis buffer containing 250 mM Imidazol as elution buffer. Subsequently, the NiNTA elution fraction was loaded on a Glutathion-sepharose column (GE-Healthcare) and eluted with lysis buffer containing 20 mM reduced Glutathion. Size exclusion chromatography

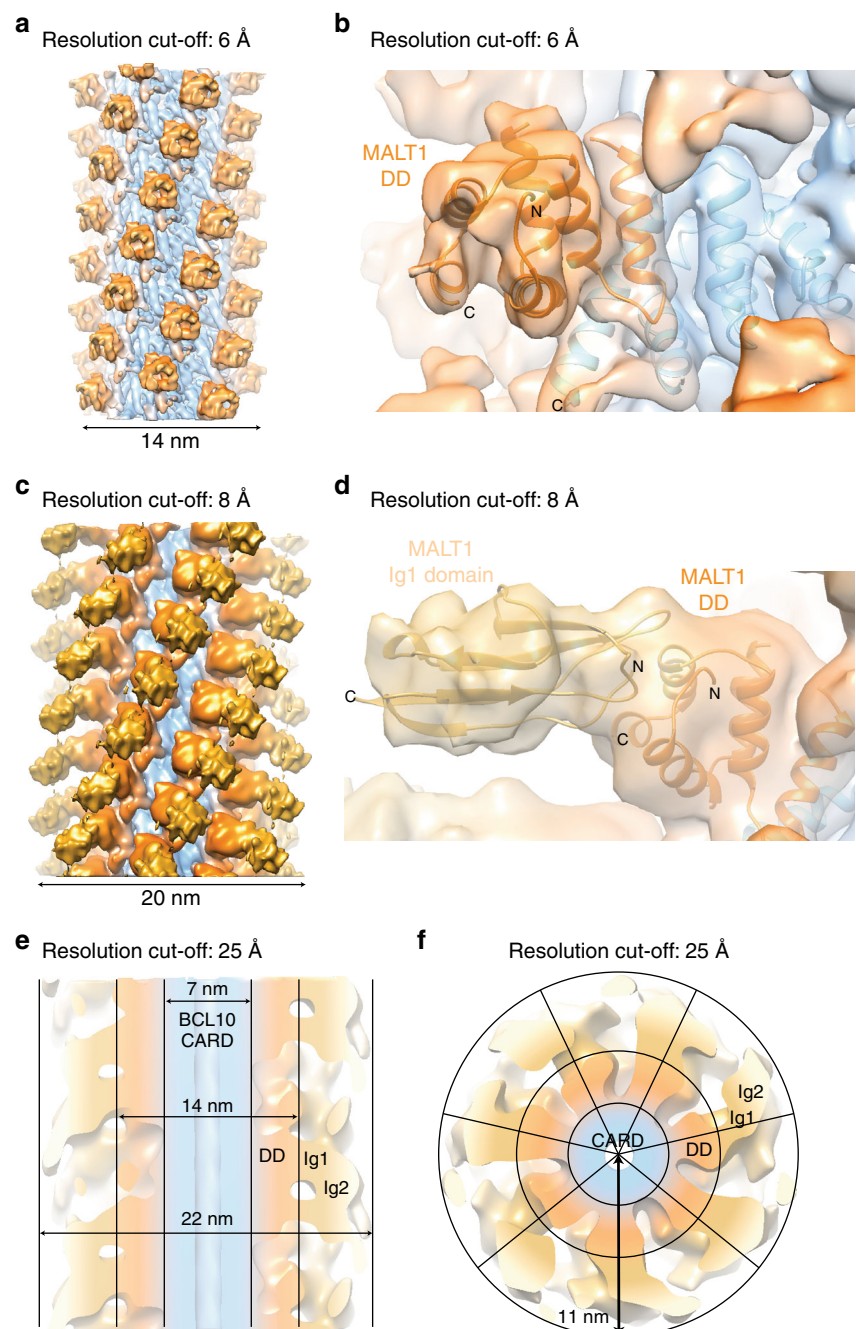

**Fig. 6** Structural organisation of additional MALT1 domains in framework of the BCL10-MALT1 filament. **a**, **b** EM density map of the well-ordered inner core of the BCL10-MALT1 filament shown together with the atomic model of BCL10 (blue) and MALT1 DD (orange) as built in the 4.9 Å density map (map sharpening B-factor applied: −200 Å$^2$). **c**, **d** Reconstructed density map filtered to 8 Å indicates the orientation of the MALT1 Ig1 domain shown as ribbon and colored yellow (B-factor applied: −100 Å$^2$). **e**, **f** Paddle-wheel like architecture of the BCL10-MALT1 filament revealed by the EM-density filtered to 25 Å (B-factor applied: 0 Å$^2$)

(S200 26/60, GE-Healthcare) was performed to separate BCL10-MALT1 complex fractions. Size exclusion buffer contained 25 mM Hepes pH = 7.5, 200 mM NaCl and 2 mM β-mercaptoethanol.

**MALT1 activity assay**. The MALT1 cleavage assay was performed with 500 nM or 10 μM of BCL10-MALT1 filaments, GST-BCL10-MALT1 species, and MALT1 V81R mutant. Protein samples were incubated in 384-well non-binding microplates and 20 μM of the MALT1 substrate Ac-LRSR-AMC was added. After 2 min incubation at 30 °C, the fluorescence of cleaved AMC was measured over 1 h by a Synergy 2 Microplate Reader (Biotek, US). The MALT1 protease activity is represented in relative fluorescence units and data from at least three independent experiments were used.

**BCL10-MALT1 complex formation analysis**. The BCL10-MALT1 complex fractions were pooled and concentrated to 0.5 μM. Filament formation was initialized by addition of PreScission protease (GE Healthcare) in equimolar amounts. Polymerization of BCL10-MALT1 species was monitored by dynamic light scattering (Viscotek 802 DLS) as a function of time over 100 min. Higher order species (above $10^4$ nm) were excluded from data evaluation. Data was analyzed by the OmniSIZE 3.0 software package (Viscotek).

**BCL10-MALT1 complex preparation for cryo electron microscopy**. BCL10-MALT1 complex fractions were concentrated stepwise to 1 mg/ml. Higher molecular species formation was monitored for each concentration step by dynamic light scattering (Viscotek 802 DLS). Filament formation was initialized by addition of equimolar amounts of  PreScission Protease (GE Healthcare) and performed

overnight at 12 °C. Analytical size exclusion chromatography (S200 5/150 GL) was performed to separate BCL10-MALT1 filaments from non-oligomerized protein. Directly before blotting Octyl-beta-Glucoside was added to the selected size exclusion fraction with a final concentration of 0.001%. 4.5 µl of the BCL10-MALT1 sample was applied on glow-discharged R1/2 grids (Quantifoil Cu R1/2, 300 mesh). The grids were blotted for 3 s at 95 % humidity and 15 °C and plunge-frozen in liquid ethane using a Leica Plunger (FEI).

**Cryo electron microscopy data collection**. Cryo-EM images of the BCL10-MALT1 complex were collected on a 200 keV Talos Arctica microscope equipped with a Falcon III detector (FEI). In total 662 micrographs were collected automatically using the EPU software (FEI). Micrographs were acquired in nano-probe mode using 49 movie frames with a dose of ~2 electrons/Å$^2$/frame resulting in a total dose of 98 electrons per Å$^2$ at a pixel size of 1.002 Å.

**Helical reconstruction**. In order to obtain both a higher resolution map of the core of the BCL10-MALT1 filaments, and a lower resolution map of the entire assembly, we performed two 3D refinements, varying the diameter limitation used during refinement, and the total dose used prior to particle extraction. For the higher resolution map, the frames 2 to 7 (total dose ~14 electrons/Å$^2$) out of a total of 49 frames were motion-corrected and dose-weighted using MotionCor2[46], while for the lower-resolution map, the frames 2 to 16 (total dose ~32 electrons/Å$^2$) were processed. The low dose dataset was used to determine the helical symmetry and to obtain the map of the BCL10-MALT1DD filament core, while the higher dose dataset enabled to get information about the periphery of the filaments.

For both datasets, the defocus estimation was performed with CTFFIND/CTFTILT[47]. In total, 2618 straight sections of filaments were boxed manually using the e2helixboxer submodule of EMAN2[48] from a selected subset of 370 micrographs. The average length of picked BCL10-MALT1 filament sections was ~756 Å and the total length ~0.2 mm (Supplementary Table 1). All subsequent processing steps were performed in the helical reconstruction software package SPRING[49]. In total 25,576 segments were extracted using a segment size of 500 Å and segment step size of 30 Å. For 2D classification and initial symmetry estimation, phase-flipped, verticalized segments were extracted from the low dose dataset, whereas for symmetry refinement and 3D reconstruction, convolved, non-rotated segments were used. The sum of the power-spectra of all verticalized segments was calculated using Segmentexam module, confirming the helical nature of the specimen. Helical parameters were determined as follows. First, 40 class averages were obtained using k-means clustering algorithm from SPARX[50] as implemented in the Segmentclass module. Class averages showed a repetition of the projection pattern along the helical axis every ~126 Å, suggesting that this distance was a close estimate of the repeat **c**. The sum of the power spectra of 2D class averages padded in 1080*1080 pixels boxes and showing the biggest number of layer lines (11 class-averages) was then calculated for symmetry estimation. Layer lines are positioned at multiples of ~1/126 Å$^{-1}$, confirming the initial estimation of the repeat (Supplementary Fig. 7a). A strong layer line ($l = 7$) with a first intensity maximum near the meridian (Bessel order **n** = 1) was attributed to the pitch **P**, with a height of 1/18 Å$^{-1}$. Worthy of note, other CARD domain assemblies have their layer line corresponding to the pitch at position $l = 9$[51]. Therefore, the structure repeats after 7 helix turns and the possible solutions of the number of units **s** are such that $s*7 = u$; **u** being the integer corresponding to the number of units in the repeat **c**. Given a range of **s** between 1.5 and 4.5, as estimated from the size of the subunit, the pitch, and the reported symmetries for CARD assemblies in the literature, we defined a range of possible values for **u** of 10 to 32. For each **u** in this range, we calculated all ratios $s = u/7$, excluding the solutions with **s** integer, as those solutions would correspond to a helix with a repeat **c** equal to **P**. Retained solutions have layer line heights matching those observed in the experimental power spectra, but have different predicted Bessel orders **n** for each layer line. The compatibility between predicted **n**'s and experimental meridional distance of first intensity maximum along layer lines in the sum of power spectra was then assessed, to narrow down the number of possible symmetries. These symmetries were further used to generate low-resolution 3D models from the 2D class-average with most signal in its power spectrum (Supplementary Fig. 7b), with the module Segclassreconstruct. Only one symmetry, with **s** = 3.571 units/turn, gave a plausible model in terms of density distribution (Supplementary Fig. 7c). This symmetry and the corresponding[47] low-resolution model were used as input for 3D refinement using Segmentrefine3D, with a limitation of diameter of 210 Å applied on the reference model at each iteration, giving an initial map with an average resolution of 5.7 Å. Comparison with existing CARD domain atomic models showed that the helix was left-handed. Importantly, when a cylinder was used as initial model instead of the model constructed from the class-average, the refinement couldn't converge to a correct 3D structure. The helical symmetry was further refined using segrefine3dgrid module, still using a diameter of 210 Å, to a pitch of 18.15 Å and 3.571 subunits per turn, corresponding to a helical rise of 5.083 Å and a helical twist of −100.81°.

Using the refined symmetry parameters and a strong segment selection based on geometrical restraints[49,52] such as calculated filament straightness (60% of straightest filaments kept) and forward x-shift difference (limited to 6 Å), the final map of the filament core was obtained using a refinement diameter of 210 Å, with 8591 segments corresponding to 51,546 asymmetric units after symmetrization.

The overall resolution of this map estimated at Fourier shell correlation (FSC) cutoff of 0.143 was 4.9 Å (Supplementary Table 1, Supplementary Fig. 8). This map was used as an initial model for the processing of the higher dose dataset using an analogous approach, but this time with the symmetry parameters were kept fixed and a higher refinement diameter of 290 Å was used for visualization of the less ordered filament exterior. Selected 8412 segments corresponding to 50,472 asymmetric units after symmetrization resulted in a map of the whole BLC10-MALT1 assembly with an overall resolution of 5.9 Å.

Both 4.9 Å and 5.9 Å resolution raw half maps were symmetrized and used to estimate the local resolution (Supplementary Fig. 8b) and the B-factor using relion_postprocess. Additionally, in order to assess the resolution of each region of the maps (Bcl10, MALT1 DD, MALT1 Ig1-Ig2-paracaspase domains) FSC curves were calculated between half maps within smooth cylinder masks of various diameters of 0–83 Å, 83–140 Å, and 140–260 Å (Supplementary Fig. 8a). The final higher resolution map of filament core (Fig. 1) used for model building was post-processed using 3Dinspect module in Spring with a b-factor of −200 Å$^{-2}$ and a resolution cutoff of 4.9 Å, whereas the lower resolution map of the entire assembly was post-processed with various resolution cut-off and sharpening, as indicated in Fig. 4.

**Model building and refinement**. The BCL10 CARD domains and MALT1 DD were built using Coot[53]. To this end, BCL10 was generated by flexible fitting of a homology model created by the program Phyre2[54] and under consideration of the BCL10 CARD domain NMR structure (pdb ID: 2MB9). The MALT1 DD was built in the EM density using the related crystal structure (pdb ID: 2G7R) as well as a homology model based on the APAF CARD domain (pdb ID: 2YGS) structure prepared with Modeller[55]. The final model was refined with PHENIX real-space refinement[56]. The similarity of the refined model of BCL10 CARD with the unsharpened, unfiltered map of the filament core was assessed by FSC, after helical symmetrization of the model and calculation of a density map with a voxel size of 1.002 Å using Chimera. Similarly, the refined model and the previously deposited model PDB: 6BZE, were compared to the EMDB entry EMD-7314. Final figures were generated with UCSF Chimera and PyMol[57,58].

**Antibodies and DNA constructs**. The following antibodies were used for immunoprecipitation (IP), Western Blot (WB), and fluorescence activate cell sorting (FACS). WB was done at a dilution of 1:1000 except when otherwise stated. Anti-HA (3F1 (WB), obtained from E. Kremmer); anti-CARD11 (1D12), anti-IκBα (L35A5, FACS 1:50), anti-p-IκBα (5A5) (all Cell Signaling Technology); anti-FLAG-M2 (F3165, Sigma-Aldrich, WB 1:10000, IP 1 µl); anti-hCD2-APC (RPA-2.10, eBioscience, FACS 1:200); anti-BCL10 (EP606Y, Abcam, for endogenous BCL10); anti-BCL10 (C-17, IP 2.5 µl) and H-197 (WB for strep-tagged BCL10); anti-MALT1 (B12), anti-β-Actin (C4; WB 1:10000), anti-CYLD (E10) (all Santa Cruz Biotechnology); anti-HOIL-1 (S150D, MRC PPU Reagents); Regnase-1 (604421, R&D; WB 1:500); horseradish peroxidase (HRP)-conjugated secondary antibodies (Jackson ImmunoResearch); anti-mouse IgG1-FITC (A85-1, BD, FACS 1:100), anti-IL-2-APC (JES6-5H4, eBioscience; FACS 1:100) and anti-Thy1.1-APC-Cy7 (HIS51, eBioscience; FACS 1:200) were used. For Jurkat T-cell stimulation anti-human CD3 and CD28 from mouse were used in the presence of anti-murine IgG1 and IgG2a (all BD Pharmingen). For CD4 T-cell stimulation anti-murine CD3 and CD28 (BD Pharmingen) from hamster were used in the presence of plate-bound rabbit anti-hamster (Jackson ImmunoResearch). Oligonucleotides for EMSAs were: H2K (fw: 5′-GATCCAGGGCTGGGGATTCCCCATCTCCACAGG-3′, rev: 5′-GATCCCTGTGGAGATGGGGAATCCCCAGCCCTG-3′), OCT1 (fw: 5′-GATCTGTCGAATGCAAATCACTAGAA-3′, rev: 5′-GATCTTCTAGT-GATTTGCATTCGACA-3′). The following DNA constructs were used: 3xFlag and HA tagged MALT1B and BCL10 cDNAs were cloned in the pEF backbone vector (Invitrogen) or pHAGE-ΔCD2-T2A (lentiviral transduction)[59] or pMSCV-IRES-Thy1.1 (retroviral transduction)[4].

**Cell culture and stimulation**. Cells were grown in DMEM (HEK293, HEK293T (both DSMZ), Phoenix-ECO (ATCC) or RPMI 1640 (Jurkat T-cells; verified by DSMZ) medium supplemented with 10% fetal calf serum and 100 U/ml penicillin/streptomycin (P/S) (all Life Technologies). HEK293 cells were transfected using standard calcium phosphate precipitation protocols. For P/I stimulation of Jurkat T-cells, 200 ng/ml PMA and 300 ng/ml Ionomycin were applied. For CD3/CD28 co-ligation, 0.3 µg anti-CD3 and 1 µg anti-CD28 antibody was used in the presence of 0.5 µg rat anti-mouse IgG1 and IgG2a.

**Generation and reconstitution of KO Jurkat T-cells**. Bicistronic expression vector px458 expressing Cas9 and sgRNA[60,61] was digested with BbsI and the linearized vector was gel purified. Targeting oligonucleotides and generation of MALT1-deficient Jurkat T-cells has been described[38]. For BCL10 two oligonucleotides targeting sites and flanking exon1 (5′AGTGAGGTCCTCCTCGGTGA 3′/5′TTCCGCTTTCGTCTCCCGCT 3′ (Supplementary Fig. 4a) were cloned. Jurkat T-cells ($4 \times 10^6$) were electroporated (220 V and 1000 µF) using a Gene pulser X (Biorad) with px458 plasmids (Addgene #48138; gift F. Zhang) containing sgRNA targeting BCL10 and EGFP expression cassette. Twenty-four hours after electroporation, EGFP-positive cells were sorted using a MoFlo sorting system (Beckman

Coulter). Isolation of clonal cell lines was achieved by serial dilutions and was followed by an appropriate expansion period. KO cell clones were initially identified by anti-BCL10 staining by Western Blot. Clones lacking protein expression were genotyped by genomic PCR using intronic primers flanking targeting sides.

For reconstitution, MALT1 isoform B and BCL10 cDNAs were linked to a C-terminal Flag-Strep-Strep tag and hΔCD2 by a co-translational processing site T2A[59] and introduced into pHAGE lentiviral expression plasmids. Lentivirus was produced by transfecting HEK293T cells with 1.5 μg psPAX2 (Addgene #12260; gift D. Trono), 1 μg pMD2.G (Addgene #12259; gift D. Trono) and 2 μg transfer vector. Seventy-two hours after transfection, lentivirus encoding MALT1 and BCL10 were collected, filtrated and in the presence of 8 μg/ml polybrene added to MALT1 and BCL10 KO Jurkat T-cells ($5 \times 10^5$ cells), respectively. After 24 h, virus was replaced by RPMI medium. FACS analysis using an Attune Flow Cytometer (Applied Biosystems) revealed >95% ΔCD2-positive cells after one week in culture.

**Purification and analysis of murine MALT1$^{-/-}$ CD4+T-cells**. CD4+ T-cells were MACS-purified from spleen and lymph nodes of MALT1$^{-/-}$ mice (MALT1$^{tm1a(EUCOMM)Hmgu}$; ES cell clone HEPD0671_C08) and stimulated with plate-bound anti-CD3/CD28 antibodies for 48 h. Retroviruses were produced in Phoenix cells transfected with pMSCV retroviral transfer vectors carrying human MALT1-FlagStrepII constructs and Thy1.1 (separated by internal ribosome entry site (IRES) sequence) and viruses were collected after 48 and 72 h. CD4+ T-cells were incubated for 6 h with retroviral supernatant supplemented with Polybrene (8 μg/ml) and then washed and cultured in RPMI medium supplemented with IL-2 (20 U/ml),10% heat-inactivated FCS, 1% P/S, 1% NEAA (Life Technologies), 1% HEPES, 1% L-glutamine, 1% sodium pyruvate (Life Technologies) and 0.1% b-mercaptoethanol for 3 days before analysis. Infection efficiencies between 25 and 50% were achieved.

For IκBα degradation, cells were stimulated for 30 min with PMA (200 nM) and Ionomycin (300 nM), fixed with 2% paraformaldehyde and stained for Thy1.1. After permeabilization (0.1% saponine), cells were stained using mouse anti-IκBα antibody (L35A5, CST) and anti-mouse IgG1 FITC (BD). For determination of intracellular IL-2, cells were rested for 12 h and then stimulated with P/I or plate-bound anti-CD3/CD28 (0.5 μg/ml CD3 and 2 μg/ml CD28) antibodies for 5 h in the presence of Brefeldin A. After fixation and permeabilization, cells were stained with anti-IL-2 APC antibody (JES6-5H4, eBioscience). FACS analyses were performed using an Attune Flow Cytometer (Applied Biosystems) and analyzed with FlowJo Software (Treestar).

**Cell lysis and cellular binding studies**. For cellular analyses Jurkat T-cells ($2–3 \times 10^6$) were lysed in co-immunoprecipitations (co-IP) buffer (25 mM HEPES (pH 7.5), 150 mM NaCl, 0.2% NP-40, 10% glycerol, 1 mM DTT, 10 mM sodium fluoride, 8 mM β-glycerophosphate, 300 μM sodium vanadate and protease inhibitor cocktail). For monitoring CBM complex formation after IP or StrepTactin pulldown (ST-PD), Jurkat T-cells ($1–3 \times 10^7$) were lysed in co-IP buffer. Lysate controls were taken up in 4xSDS-loading buffer and boiled for 5 min at 95 °C. IP was carried out using anti-BCL10 C-17 (2.5 μl) or anti-Flag-M2 (1 μl) antibodies overnight at 4 °C. Afterwards Protein G Sepharose (15 μl 1:1 suspension) was administered for 1–2 h at 4 °C to bind antibodies. StrepII-tagged were pulled down (ST-PD) with Strep-Tactin Sepharose (15 μl 1:1 suspension) at 4 °C overnight. For co-IP and ST-PD beads were washed with co-IP buffer and boiled after adding of 22 μl 2x SDS loading buffer. Lysates and precipitated proteins were separated by SDS-PAGE and analyzed by Western blot.

**Western blotting**. Transfer onto PVDF-membranes was performed using electrophoretic semi-dry blotting system. PVDF-membranes were blocked with 5% BSA for 1 h at RT and before primary antibody (indicated above, dilution 1:1000 in 2.5% BSA/PBS-T) were incubated overnight at 4 °C. Membranes were washed in PBS-T and HRP (horse radish peroxidase)-coupled secondary antibodies (indicated above, 1:7000 in 1.25% BSA in PBS-T; 1 h, RT) were used for detection. HRP was visualized by enhanced chemiluminescence (ECL) with LumiGlo reagent (Cell Signaling Technologies) according to the protocol of the manufacturer. Images were cropped for presentation and full size images are presented in Supplementary Fig. 9.

**Electrophoretic mobility shift assay**. Electrophoretic mobility shift assays (EMSAs) were carried out using double-stranded NF-κB or OCT1 binding sequences (H2K or OCT1 oligonucleotides; see reagents) which were radioactively labeled using [α-32P] dATP in a Klenow Fragment (NEB) reaction. Whole cell lysates (3–6 μg) were incubated for 30 min at RT with shift-buffer (HEPES pH 7.9 (20 mM), KCl (120 mM), Ficoll (4%)), DTT (5 mM), BSA (10 μg) and poly-dI-dC (2 μg, Roche) and radioactive probe (10,000–20,000 cpm) to detect DNA binding of transcription factors. Samples were run on a 5% polyacrylamide gel in TBE buffer and exposed to autoradiography after vacuum-drying. Images were cropped for presentation and full size images are presented in Supplementary Fig. 9.

**Data availability**. The electron density reconstruction and final model were deposited with the EM Data Base (accession codes EMD-0013, PDB ID 6GK2). Other data are available from the corresponding authors upon reasonable request.

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

## Acknowledgements
We thank the MPI of Biochemistry cryo-EM and biophysics core facilities for generous access and support. We are grateful to the IT teams from IBS and EMBL Grenoble, and specifically Aymeric Peuch for help with the usage of the joint IBS/EMBL EM computing cluster which was used as a part of the platforms of the Grenoble Instruct-ERIC Center (ISBG: UMS 3518 CNRS-CEA-UGA-EMBL) with support from FRISBI (ANR-10-INSB-05-02) and GRAL (ANR-10-LABX-49-01) within the Grenoble Partnership for Structural Biology (PSB). We thank Simon Widmann for help with the cell-based assays. K.L. and D.K. are supported by the CRC1054 project B02 and A04, respectively. A.D. is supported by the FRM ARF20160936266 grant. K.-P.H is supported by the Deutsche Forschungsgemeinschaft (CRC1064, GRK1721), the European Research Council (ERC Advanced Grant ATMMACHINE), the Gottfried-Wilhelm-Leibniz Prize and the Center for Integrated Protein Sciences Munich (CIPSM).

## Author contributions
K.L. and F.S. prepared the cryo-EM samples and collected the data. T.S. performed all cellular experiments. A.D. performed helical reconstruction and helped with structure determination. I.G. helped with EM data analysis. K.L. built atomic models. F.S. prepared the protein complex, the biochemical analysis and participated in structure determination. T.G. established the Jurkat KO T-cells and performed reconstitution of murine MALT1$^{-/-}$ CD4 T-cells. M.S. operates the MPI Biochemistry cryo-EM facility, helped with EM data collection and provided general EM advice. K.L., D.K., and K.-P.H. designed the overall study, analyzed the results and wrote the paper with input of A.D. and I.G. and contributions from all other authors.

## Additional information

**Competing interests:** The authors declare no competing interests.

