## [Peer Review File · Nature Communications]

Reviewers' comments:

Reviewer #1 (Remarks to the Author):

Here Schlauderer and colleagues present a refined structure of the BCL10-MALT1 filament using cryo-EM. Their findings are largely congruent with elegant work from the Wu lab (references 3 and 22), with important new insights and implications that extend beyond these recent publications. Perhaps most surprisingly, their data suggests that both BCL10-MALT1 (BM-I interface) and BCL10-BCL10 interactions are required for TCR-induced CARD11 binding, full CBM complex formation, and downstream NF- κ B activation. Strong biochemical data using structure-informed mutagenesis, co-IPs and signaling readouts in T cells largely support their conclusions. These results imply that BCL10 core filament formation, along with a pre-existing MALT1 interaction, actually precedes CARD11 recruitment – a compelling concept that challenges the accepted model whereby CARD11 recruits BCL10 to nucleate filament formation. Overall, the paper is well crafted and marks a potentially significant contribution to the field.

Major critiques:

1. The most intriguing finding of the paper is that BCL10-BCL10 interactions within interface I of the core filament are actually required for CARD11 recruitment, as suggested by the BCL10 R42E IP experiment shown in Figure 2i. Although R42 is not part of the proposed CARD11-BCL10 interface between their respective CARD domains, the authors have not ruled out this possibility with a definitive experiment. Could the R42E substitution interfere with the proper positioning of the adjacent acidic interface (E50, E53 and E54) proposed to interact with the CARD11 CARD (see Li et al (2012) PLoS One 7:e42775)? Testing the R36E mutant and/or different amino acid substitutions at position 42 (e.g. R42A) in co-IPs with CARD11 could lend further credence to their claim. In silico modeling of R36E or R42E variants within their core structure (based on Ext Data Figure 2) could also be considered to demonstrate these substitutions do not significantly alter the structure of the BCL10 CARD and/or the acidic interface with CARD11.
2. It is not clear where or how the S/T rich C-terminal region of BCL10 fits into the cryo-EM / atomic model shown in Figure 1c-d.
3. BCL10 itself is a substrate of MALT1 protease at Arg228 (Ribeaud et al (2008) Nat Immunol 9:272-81). If the MALT1 paracaspase domains are facing outward away from the core BCL10 filament, how do the authors propose MALT1 can cleave the C-terminal end of BCL10?

Minor critiques:

1. The BCL10 subunits labels in Figure 2a-d (A1, B1, etc.) are not well described in the text or figure legend – the authors should clarify. Moving diagrams shown in Extended Fig 2a-c to the main figures might be helpful in this regard.
2. In discussing germline LOF mutations in CBM components that drive CID, the authors should also cite additional key references:
 - a. Stepensky et al (2013) J Allergy Clin Immunol 131:477-85 (additional CARD11-deficient patient, concomitant report with ref #11)
 - b. Torres et al (2014) J Clin Invest 124:5239-48. (BCL10 deficiency in humans).
 - c. Ma et al (2017) Nat Genet 49:1192-1201. (inherited CARD11 LOF mutations in atopic patients).

Reviewer #2 (Remarks to the Author):

Schlauderer et al. determined the cryo-EM structure of the BCL10•MALT1 complex and validated the resulting model using various cellular and in vitro methods. The overall architecture of the same complex was recently proposed by Wu and colleagues; however, they were not able to

reconstruct the BCL10•MALT1 complex. The current manuscript closes this gap by solving the cryo-EM structure of the BCL10•MALT1 complex in near-atomic resolution. Thus, the presented study marks an important extension of the recently published studies by Wu and colleagues.

Major points:

1. The claimed resolution of the BCL10 filament is 4.9Å; however, in Fig. 1h, one can easily see the side-chains. I'm a bit puzzled by why that is.
2. The major novelty of the present study is the identification of BCL10•MALT1 interfaces via cryo-EM, which the authors used mutagenesis studies to test their implications. It is clear that V81 plays an important role, supporting the cryo-EM model. However, the authors kept V81R with other mutations (L82D/E75A, both of which are located in different interfaces). It appears that the V81R mutant acted as a dominant negative. In order to validate the cryo-EM model, the authors need to generate these other mutations individually and test whether they still affect the complex formation.
3. The authors mentioned that helical reconstruction using a featureless cylinder did not work. Moreover, the 2D class average resembled a corkscrew. I am curious as to whether the approach the authors used might have biased the model building. For instance, the binding of MALT1 to BCL10 might have changed the overall contour of filament.

Minor point:

1. The labels in Fig. 3A are confusing. There are four lanes in the gel, but only three labels.

Reviewer #3 (Remarks to the Author):

This cryo-EM analysis of BCL-MALT1 filaments by Schlauderer et al takes a deep dive into the two-component filament involved in linking immunogenic signals to an intracellular immune response by C- and T-cells. The structure is of high quality and accompanied by minimal biochemical but fairly comprehensive cell biological analysis to validate the interpretation of the structure. The researchers show that the helical assembly becomes decreasingly rigid at higher diameters, moving out from the BCL10 core to the multidomain MALT1. MALT 1 shows varying resolutions for each of the domains that are linked by flexible regions, from the death domain that binds BCL10 to the first two Ig motifs. The procaspase and third Ig domains are not ordered but the purified specimen has protease activity suggesting they are intact in the specimen.

The manuscript was a jumble of alphabet soup. I realize the authors cannot control the field's propensity for complicated jargon, but it would be nice to see a streamlined introduction that better explains why this minimal complex – if I understood, the helix contains only two of the three components needed for signal transduction – is important. For example, from their abstract, I expected to learn about the assembly that contains CARD11 and it doesn't seem like they can say much about that beyond that it is linked to MALT1 binding BCL10 in terms of the structure that is central to the manuscript. I found the labels in Figure 2 confusing because they used a numbering scheme to label successive BCL10 subunits along the helix (A1, A2, A3, etc) they did not explain and was not obvious until I got to Extended Data Figure 2 – and even there it had to be inferred from panel c and was not explained in the legend or discussed in the text that I could pick find.

The above issues are structural but diminish impact and significance of the results. They should be addressed with revision to the text.

I also have a question about the hairy appearance of the filaments. I was surprised they could get to high resolution because the filaments appear quite heterogeneous in the raw image and appear significantly wider than the final structure – if I am reading the scale-bar accurately, they are about 90 nm wide as raw filaments but the most of that width they can resolve even at lower

resolution is about 20 nm in diameter. Is that accurate? It seems like a lot of protein for just the capase and the third Ig domains from MALT1. Clarification of this seeming disconnection would be nice, perhaps even with some immunogold labeling to show those knobs are the C-terminal end of MALT1.

It would also be helpful to understand the impact of the knobs on the image analysis by providing a visual mark of the relative size of the narrow and wide boxes and the resulting reconstructions. How much of that knobby density is lopped off? Did the researchers need to use a tight mask to avoid interference?

The above issues can be addressed by clarifying details about the disconnection between the raw images and the final structure. I understand immunolabeling EM samples can be difficult. Nevertheless, labeling would enhance the significance of the story because it would give experimental credence to the model that MALT1's disordered C-terminal domains are important for recruiting other signaling factors.

Point-by point response to Reviewer:

Reviewers' comments:

We thank all reviewers for their positive evaluation and some very constructive comments and suggestions. We performed some further experiments and address all in detail in the point-by-point response. We would like to mention that we have modified the rather short report to a full article with now six main figures and a more detailed description in the text, including an introduction and discussion part. We are confident that this improved version is now more comprehensive also for readers that are not directly from the field.

Reviewer #1 (Remarks to the Author):

Here Schlauderer and colleagues present a refined structure of the BCL10-MALT1 filament using cryo-EM. Their findings are largely congruent with elegant work from the Wu lab (references 3 and 22), with important new insights and implications that extend beyond these recent publications. Perhaps most surprisingly, their data suggests that both BCL10-MALT1 (BM-I interface) and BCL10-BCL10 interactions are required for TCR-induced CARD11 binding, full CBM complex formation, and downstream NF- κ B activation. Strong biochemical data using structure-informed mutagenesis, co-IPs and signaling readouts in T cells largely support their conclusions. These results imply that BCL10 core filament formation, along with a pre-existing MALT1 interaction, actually precedes CARD11 recruitment – a compelling concept that challenges the accepted model whereby CARD11 recruits BCL10 to nucleate filament formation. Overall, the paper is well crafted and marks a potentially significant contribution to the field.

Major critiques:

1. The most intriguing finding of the paper is that BCL10-BCL10 interactions within interface I of the core filament are actually required for CARD11 recruitment, as suggested by the BCL10 R42E IP experiment shown in Figure 2i. Although R42 is not part of the proposed CARD11-BCL10 interface between their respective CARD domains, the authors have not ruled out this possibility with a definitive experiment. Could the R42E substitution interfere with the proper positioning of the adjacent acidic interface (E50, E53 and E54) proposed to interact with the CARD11 CARD (see Li et al (2012) PLoS One 7:e42775)? Testing the R36E mutant and/or different amino acid substitutions at position 42 (e.g. R42A) in co-IPs with CARD11 could lend further credence to their claim. In silico modeling of R36E or R42E variants within their core structure (based on Ext Data Figure 2) could also be considered to demonstrate these substitutions do not significantly alter the structure of the BCL10 CARD and/or the acidic interface with CARD11.

We chose BCL10 R42E in the interface I, because it is pointing away from the α 2 helix and not involved in maintaining the CARD conformation in a BCL10 monomer. However, we acknowledge that it is possible that change of a positively charged arginine to a negatively charged glutamic acid may cause additional conformational alterations. As suggested by the reviewer we tested the milder R42A

substitution and also the R36E substitution in the interface II. Both substitutions show the same effect as the BCL10 R42E mutation, confirming that BCL10 filament assembly is required for its recruitment to CARD11 (new Figure 3). In fact, none of these mutants prevents MALT1 binding, suggesting that the overall conformation of the BCL10 CARD is unchanged. In line, all BCL10 interface mutations abolish P/I-induced NF- κ B signaling and MALT1 protease activation. Also, we would like to point out that in our view it is striking, but not so surprising that BCL10 oligomerization is required for CARD11 recruitment. Assembly of large signaling clusters often relies on the additive effect of multiple low affinity interactions. In the case of BCL10, we think that a monomer may weakly bind to a multimeric CARD11 seed, but without the combined affinity of the BCL10 oligomer, this is not sufficient for a stable CARD11-BCL10 complex. Similar, MALT1 may stabilize the BCL10 CARD to allow efficient recruitment to CARD11.

To make these important points clearer, we decided to move the structure of the BCL10-BCL10 interfaces and the functional analyses in Figure 2 and 3. In addition, we have taken it up in the Discussion in the second paragraph on page 9-10.

2. It is not clear where or how the S/T rich C-terminal region of BCL10 fits into the cryo-EM / atomic model shown in Figure 1c-d.

The BCL10 C-terminus is most likely flexible and was not visible in the cryo-EM map, nevertheless between the MALT1 molecules would be enough space to accommodate the C-terminal residues.

3. BCL10 itself is a substrate of MALT1 protease at Arg228 (Ribeaud et al (2008) Nat Immunol 9:272-81). If the MALT1 paracaspase domains are facing outward away from the core BCL10 filament, how do the authors propose MALT1 can cleave the C-terminal end of BCL10?

BCL10 encodes for 233 amino acids and it is well possible that the R228 is positioned in a way that it could be cleaved within the BCL10-MALT1 filament. As pointed out above, we unfortunately do not see the flexible BCL10 C-terminus. Other substrates must be recruited to MALT1 paracaspase from outside the filament and it remains possible that BCL10 proteins are cleaved, which are not part of the filaments. In fact, since BCL10 cleavage has been associated with integrin signaling independent of CARD11 and MALT1, this could be an attractive option. However, our structure does not provide any rationale for this hypothesis and thus we do not want to speculate about it.

Minor critiques:

1. The BCL10 subunits labels in Figure 2a-d (A1, B1, etc.) are not well described in the text or figure legend – the authors should clarify. Moving diagrams shown in Extended Fig 2a-c to the main figures might be helpful in this regard.

Now, we provide the BCL10 core filament description and the BCL10 interface figure in the main text (new Fig. 2). Given the detailed biochemical and functional analyses of the BCL10 filaments in BCL10 KO cells and the new insights about the CBM assembly, we moved the BCL10 cell analyses (new Figure 3) to the main part too.

2. In discussing germline LOF mutations in CBM components that drive CID, the authors should also cite additional key references:

- a. Stepensky et al (2013) J Allergy Clin Immunol 131:477-85 (additional CARD11-deficient patient, concomitant report with ref #11)
- b. Torres et al (2014) J Clin Invest 124:5239-48. (BCL10 deficiency in humans).
- c. Ma et al (2017) Nat Genet 49:1192-1201. (inherited CARD11 LOF mutations in atopic patients).

We included the above-mentioned references in the introduction section.

Reviewer #2 (Remarks to the Author):

Schlauderer et al. determined the cryo-EM structure of the BCL10•MALT1 complex and validated the resulting model using various cellular and in vitro methods. The overall architecture of the same complex was recently proposed by Wu and colleagues; however, they were not able to reconstruct the BCL10•MALT1 complex. The current manuscript closes this gap by solving the cryo-EM structure of the BCL10•MALT1 complex in near-atomic resolution. Thus, the presented study marks an important extension of the recently published studies by Wu and colleagues.

Major points:

1. The claimed resolution of the BCL10 filament is 4.9Å; however, in Fig. 1h, one can easily see the side-chains. I'm a bit puzzled by why that is.

In the best resolved parts of the cryo-EM map, filtered to 4.9 Å for visualization, one can clearly see certain side chains. The most rigid, most bulky and less radiation damage-sensitive side chains are indeed visible in cryo-EM maps even at this resolution.

2. The major novelty of the present study is the identification of BCL10•MALT1 interfaces via cryo-EM, which the authors used mutagenesis studies to test their implications. It is clear that V81 plays an important role, supporting the cryo-EM model. However, the authors kept V81R with other mutations (L82D/E75A, both of which are located in different interfaces). It appears that the V81R mutant acted as a dominant negative. In order to validate the cryo-EM model, the authors need to generate these other mutations individually and test whether they still affect the complex formation.

V81R, L82D and E75A are all in the same interface that we define as BM-I and that connects the MALT1 DD $\alpha 5$ helix with the BCL10 CARD $\alpha 5$ and $\alpha 6$ helices. Initially, we designed the single V81R and the triple (V81R/L82D/E75A), because we were not sure if a single mutation would be sufficient to abolish interaction. However, we agree that having a complete loss of interaction with V81R, it does not make sense to include the triple mutant and we removed it. Instead, to get an independent verification for the critical requirement of the interface, we mutated and expressed MALT1 L82D in MALT1 KO Jurkat T-cells. Again, this mutant led to loss of BCL10-MALT1 interaction, severely reduced recruitment of BCL10 to CARD11 and NF- κ B signaling as well as MALT1 protease activation was abolished (new Figures

5c and h). We want to emphasize that reduced CARD11-BCL10 binding is also seen in MALT1 KO Jurkat T-cells and after mutation of the MALT1 opposing surface on BCL10 (L104R). Thus, all together the data provide clear evidence that BCL10-MALT1 association is required for CBM complex formation and we discuss this important finding on page 10 (first paragraph).

We do not think and do not claim that MALT1 BM-I mutations have dominant negative effects. These are loss of function mutations and all experiments were done in clean genetic setting using reconstitution of MALT1 deficient Jurkat T-cells that were initially described in Meininger et al (Nat. Commun 2016; 7, 11292.). Thus, no endogenous MALT1 is present. We think that this is a strength regarding the biochemical analyses of CBM complex formation and the functional effects on signaling in T-cells.

3. The authors mentioned that helical reconstruction using a featureless cylinder did not work. Moreover, the 2D class average resembled a corkscrew. I am curious as to whether the approach the authors used might have biased the model building. For instance, the binding of MALT1 to BCL10 might have changed the overall contour of filament.

The fact that correct helical parameters cannot be found when using a featureless cylinder as an initial model is very common in the helical analysis field, because refinement of noisy data with an inappropriate reference can be easily trapped in a local minimum. This is precisely why the indexing based on layer lines, much more rigorous and unbiased than a simple reference-based refinement, although much more labor-intensive, is still required in difficult cases. The success of the indexing procedure in the case of the BCL10-MALT1 complex demonstrates the relatively high rigidity of these filaments. Moreover, the helical parameters derived from the indexing of the BCL10-MALT1 (helical rise of 5.083 Å and a helical twist of -100.81°) are virtually identical to those determined for BCL10 alone by Wu and colleagues (helical rise of 5.00 Å and a helical twist of -100.8°), which in itself is the best proof of the absence of any effect of our method on the overall contour of the filaments.

Minor point:

1. The labels in Fig. 3A are confusing. There are four lanes in the gel, but only three labels.

We have added minus to indicate that P/I was not added in the control lanes (new Figures 3c and d).

Reviewer #3 (Remarks to the Author):

This cryo-EM analysis of BCL-MALT1 filaments by Schlauderer et al takes a deep dive into the two-component filament involved in linking immunogenic signals to an intracellular immune response by C- and T-cells. The structure is of high quality and accompanied by minimal biochemical but fairly comprehensive cell biological analysis to validate the interpretation of the structure. The researchers show that the helical assembly becomes decreasingly rigid at higher diameters, moving out from the BCL10 core to the multidomain MALT1. MALT 1 shows varying resolutions for each of the domains that are linked by flexible regions, from the death domain that binds BCL10 to the first two Ig motifs. The

procaspase and third Ig domains are not ordered but the purified specimen has protease activity suggesting they are intact in the specimen.

The manuscript was a jumble of alphabet soup. I realize the authors cannot control the field's propensity for complicated jargon, but it would be nice to see a streamlined introduction that better explains why this minimal complex – if I understood, the helix contains only two of the three components needed for signal transduction – is important. For example, from their abstract, I expected to learn about the assembly that contains CARD11 and it doesn't seem like they can say much about that beyond that it is linked to MALT1 binding BCL10 in terms of the structure that is central to the manuscript. I found the labels in Figure 2 confusing because they used a numbering scheme to label successive BCL10 subunits along the helix (A1, A2, A3, etc) they did not explain and was not obvious until I got to Extended Data Figure 2 – and even there it had to be inferred from panel c and was not explained in the legend or discussed in the text that I could pick find.

We apologize that the short format has caused some confusion and we are confident that the extended version of the full article is more comprehensive. The fact that the manuscript resolves the BCL10-MALT1 structure is actually in the title and we think that this should be clear. In the introduction we now describe in more details the components of the CBM complex and what is functionally and structurally known. Whereas BCL10 filaments were solved and the CARD11 seed function has been addressed by modeling from the Wu lab, we now add how the essential regulatory subunit MALT1 connects to the BCL10 filaments. The structural data are confirmed by comprehensive biochemical and functional analyses.

The above issues are structural but diminish impact and significance of the results. They should be addressed with revision to the text.

I also have a question about the hairy appearance of the filaments. "I was surprised they could get to high resolution because the filaments appear quite heterogeneous in the raw image and appear significantly wider than the final structure – if I am reading the scale-bar accurately, they are about 90 nm wide as raw filaments but the most of that width they can resolve even at lower resolution is about 20 nm in diameter. Is that accurate? It seems like a lot of protein for just the caspase and the third Ig domains from MALT1. Clarification of this seeming disconnection would be nice, perhaps even with some immunogold labeling to show those knobs are the C-terminal end of MALT1.

We are sorry for this confusion that probably arose from the fact that the scale bar at the Figure 1b was represented as a three-segment black-white-black line of 90 nm in total. Each scale bar segment of 30 nm in length was meant to illustrate that that BCL10-MALT1 assembles into flexible helical filaments of ~29 nm in diameter as specified in the main text. To avoid this misunderstanding, we now show the scale bar at the Figure 1b as a one-color 30 nm line and indicated in Fig. 1c a diameter of 29nm.

It would also be helpful to understand the impact of the knobs on the image analysis by providing a visual mark of the relative size of the narrow and wide boxes and the resulting reconstructions.

This information was provided in Figure 1. We now also provide the required visual marks at the Figure 1b-c. Actually, dimensions of different parts of the structure were also shown at the cut-through view at the Figure 6e. The increase of the flexibility towards the filament's periphery can be appreciated also from the Supplementary Data Figure 8.

How much of that knobby density is lopped off? Did the researchers need to use a tight mask to avoid interference?

As clarified above, the reviewer was certainly misled by the appearance of the scale bar at the Figure 1b. No peripheral density was excluded in our calculations. The 4.9 Å resolution map of the filament core was obtained using a refinement diameter of 210 Å, and used as an initial model for refinement of the 290 Å diameter map to allow visualisation of the less ordered filament exterior.

The above issues can be addressed by clarifying details about the disconnection between the raw images and the final structure. I understand immunolabeling EM samples can be difficult. Nevertheless, labeling would enhance the significance of the story because it would give experimental credence to the model that MALT1's disordered C-terminal domains are important for recruiting other signaling factors.

We hope that the explanations that we provide above clear up the unfortunate misunderstanding of the filament dimensions, thereby removing the necessity of immunolabelling.

REVIEWERS' COMMENTS:

Reviewer #1 (Remarks to the Author):

The authors have done a great job in thoroughly addressing my critiques as well as comments from the other Reviewers. The result is an excellent, expanded full article that provides novel insights into CBM complex assembly and advances the field.

Reviewer #2 (Remarks to the Author):

The authors have responded to criticisms carefully and have added important new data. I have no further comments.